# The Big Five Personality Traits as Predictors of Negative Emotional States in University Students in Taiwan

**DOI:** 10.3390/ijerph192416468

**Published:** 2022-12-08

**Authors:** Shih-Wei Yang, Malcolm Koo

**Affiliations:** 1Department of Education, National Chiayi University, Chiayi City 600355, Taiwan; 2Department of Styling and Cosmetology, Tainan University of Technology, Tainan City 710302, Taiwan; 3Graduate Institute of Long-Term Care, Tzu Chi University of Science and Technology, Hualien City 970302, Taiwan; 4Dalla Lana School of Public Health, University of Toronto, Toronto, ON M5T 3M7, Canada

**Keywords:** emotion, stress, anxiety, depression, personality, students

## Abstract

Negative emotional states, such as stress, anxiety, and depression, are prevalent in university students. Personality traits have been shown to be associated with a wide range of behaviors in students, such as academic motivation, achievement, and social well-being. The aim of this study was to investigate the association between the Big Five personality traits and negative emotion states in university students in Taiwan. A cross-sectional study was conducted on 580 university students in Taiwan. Negative emotional states were evaluated using the Depression Anxiety Stress Scale-21 (DASS-21) and the Big Five personality traits were measured using the 48-item Big Five Inventory. A hierarchical linear regression analysis was used to assess the factors associated with DASS-21 scores. Neuroticism (standardized beta [std. β] = 0.45, *p* < 0.001) and openness (std. β = 0.12, *p* = 0.003) were significantly associated with DASS-21 scores, while agreeableness (std. β = −0.10, *p* = 0.007) was significantly and inversely associated with DASS-21 scores. Personality traits could be used to identify students at risk of negative emotional states and to undertake appropriate preventive strategies.

## 1. Introduction

Mental health is an important component of overall health at all stages of life [1]. Negative psychological well-being indicators, such as stress, anxiety, and depression, have been found to be increasing in the past decades around the world [2]. As university life can be stressful for students as they go through a transitory period from adolescence to adulthood [3], the mental health of university students is an area of particular concern. A study of 1617 university students in Turkey showed a high prevalence of depression (27.1%), anxiety (47.1%), and stress (27%) [4]. A cross-sectional study of 1074 Spanish college students also reported a prevalence of 18.4%, 23.6%, and 34.5% for depression, anxiety, and stress, respectively [5]. A cohort study of 1686 undergraduate students in the United Kingdom found that 32% of students reported moderate to severe anxiety symptoms and 27% of students reported moderate to severe depressive symptoms at entry to university [6]. In addition, multiple studies in the United States have suggested that there was an increasing prevalence of depression and anxiety among university students [7,8]. Furthermore, a web-based survey of 7915 first-year tertiary education students in Hong Kong found that 21%, 41%, and 27% of the respondents had moderate or more severe levels of depression, anxiety, and stress, respectively [9]. A systematic review and meta-analysis of 113 studies with 185,787 Chinese university students revealed that the overall prevalence of depression was 28.4% [10]. These findings highlight the need to address mental health in university students.

The five-factor model of personality proposes that personality attributes can be comprehensively grouped along five basic dimensions, namely, neuroticism, extraversion, openness to experience, agreeableness, and conscientiousness [11]. It has been associated with a wide range of behaviors in students, such as academic achievement [12], learning strategies [13], social well-being [14], and mental health [15]. Previous research has also demonstrated a number of associations between mental health and the five-factor model of personality in adults [16,17]. Similar associations were also observed in adolescents and students. For example, a higher level of neuroticism is a risk-factor for depression and suicidal behavior [18,19]. A cross-sectional study of 323 Chinese undergraduates reported that neuroticism, conscientiousness, and agreeableness significantly predicted anxiety among college students [20]. A study of 1744 students studying veterinary medicine, medicine, dentistry, pharmacy, and law in the United Kingdom showed that high levels of neuroticism and low conscientiousness were risk factors for increased psychological morbidity [21]. A six-year longitudinal study in Norway identified that a combination of high neuroticism, high conscientiousness, and low extraversion could predict medical school stress [22]. A review study of 66 meta-analyses with 851 effect sizes revealed that while neuroticism was consistently associated with common mental disorders, other traits also showed substantial independent effects [18]. Nevertheless, few studies have explored the relationship between personality traits and negative emotional states among university students in Taiwan. Therefore, the aim of the present study was to investigate the association between negative emotional states and the Big Five personality traits in a sample of university students in Taiwan.

## 2. Materials and Methods

### 2.1. Study Design and Participants

This cross-sectional study recruited students from two universities in southern Taiwan and two universities of science and technology in southern and eastern Taiwan. The inclusion criteria included students who were full-time and between the age of 20 and 26 years. Class mentors at the selected universities were contacted by our research assistants for permission to conduct a survey during the regular class meeting time.

### 2.2. Data Collection

The study protocol was approved by the Research Ethnics Committee of Hualien Tzu Chi Hospital, Buddhist Tzu Chi Medical Foundation (No. IRB109-252-B). The study was carried out according to the Declaration of Helsinki. Respondents were informed that participation was voluntary. Only those who provided their informed consent were enrolled in the study.

Anonymous paper-based questionnaires were distributed by research assistants, between 24 November 2021 and 26 October 2022, to students who agreed to participate in the study. Upon completion of the questionnaire, respondents would receive a convenient store gift card valued at 100 New Taiwan Dollars (approximately US$3) as an honorarium.

### 2.3. Measurements

#### 2.3.1. Background Information of the Participants

A self-administered paper-based questionnaire was used to assess demographic data, Big Five personality traits, and negative emotional states of the respondents. Demographic information included sex, age, body weight, body height, study program, type of institution, and self-perceived health status. Body mass index was calculated by dividing a person’s weight in kilograms by the square of their height in meters. It was further categorized into four groups: underweight (<18.5 kg/m^2^), normal (≥18.5 kg/m^2^ and <24.0 kg/m^2^), overweight (≥24.0 kg/m^2^ and <27.0 kg/m^2^), and obesity (≥27.0 kg/m^2^) according to the cut-off points recommended by the Health Promotion Administration, Ministry of Health and Welfare of Taiwan. Body mass index was calculated in the present study because recent research has shown that it was associated with depression and psychosocial stress [23]. Self-perceived health status was measured using a single item with a five-point scale from very good to very poor.

#### 2.3.2. Measurement of Health Promoting Behaviors

Health-promoting behaviors were ascertained by the 52-item Health-Promoting Lifestyle Profile II: Chinese version (HPLP-II). The original Health Promoting Lifestyle Profile was developed by Walker et al. [24] and the HPLP-II has been translated and psychometrically validated in different populations, including Taiwanese women, with an overall Cronbach’s alpha of 0.95 and of 0.73 to 0.91 for the subscales [25]. The scale consists of six subscales, including health responsibility (9 items), physical activity (8 items), nutrition (9 items), spiritual growth (9 items), interpersonal relations (9 items), and stress management (8 items). Respondents were asked to rate the frequency with which they practiced each of the 52 behaviors on a four-point Likert-type scale ranging from 1 (never) to 4 (routinely). A higher score indicates a better health lifestyle. In this study, the internal reliability, McDonald’s omega, for the health responsibility, physical activity, nutrition, spiritual growth, interpersonal relations, and stress management subscales were 0.841 (95% confidence interval [CI] 0.814, 0.865), 0.845 (95% CI 0.818, 0.866), 0.698 (95% CI 0.644, 0.743), 0.875 (95% CI 0.855, 0.891), 0.830 (95% CI 0.808, 0.850), and 0.770 (95% CI 0.733, 0.799).

#### 2.3.3. Measurement of the Big Five Personality Traits

The Big Five personality traits were assessed using the Big Five Inventory (BFI). The BFI consists of 44 items with five scales: openness to experience (10 items), conscientiousness (9 items), extraversion (8 items), agreeableness (9 items), and neuroticism (8 items). Respondents were asked to indicate their agreement with each item using a 5-point Likert scale ranging from 1 (disagree strongly) to 5 (agree strongly) [26]. The BFI showed a clear five-factor structure and convergent validity with other Big Five scales, such as the Revised NEO Personality Inventory (NEO PI-R) [27]. Its five-factor structure has been substantially replicated in many cultures [28], including Chinese [29,30]. The internal reliability Cronbach’s alpha values for openness to experience, conscientiousness, extraversion, agreeableness, and neuroticism were 0.83, 0.82, 0.86, 0.79, and 0.87, respectively [26]. In this study, the internal reliability values, McDonald’s omega, for openness to experience, conscientiousness, extraversion, agreeableness, and neuroticism were 0.793 (95% CI 0.754, 0.821), 0.734 (95% CI 0.684, 0.770), 0.801 (95% CI 0.767, 0.827), 0.648 (95% CI 0.579, 0.703), and 0.745 (95% CI 0.703, 0.777), respectively.

#### 2.3.4. Measurement of Negative Emotional States

The main outcome variable, negative emotional states, was assessed by using the standardized Depression Anxiety Stress Scale-21 (DASS-21). The respondent was asked to rate on a scale that ranged from 0 (did not apply to me at all) to 3 (applied to me very much or most of the time) to indicate their level of agreement with the symptoms they experienced in the past week. Higher scores reflect higher levels of symptom endorsement [31].

The DASS-21 has been widely used in a range of studies from different countries with different samples with good reliability and validity [32,33]. A recent study examined the dimensionality, invariance, and reliability of the DASS-21 in 2580 college students from Brazil, Canada, Hong Kong, Romania, Taiwan, Turkey, United Arab Emirates, and the United States. The results suggested that DASS-21 could be used as a unidimensional scale to represent a general score of distress [34]. In the present study, the total score of the DASS-21 was used to represent negative emotional states. The items’ scores were added and multiplied by two to obtain the total score that could be compared with the original DASS-42 [35]. In this study, the reliability coefficient, McDonald’s omega, of the DASS-21a was excellent at 0.931 (95% CI 0.920, 0.941).

### 2.4. Statistical Analysis

Data analysis was conducted with IBM SPSS for Windows, version 28 (IBM, Armonk, NY, USA). Data were summarized using mean and standard deviation (SD) or frequency and percentage, as appropriate.

The association between DASS-21 scores and independent variables were analyzed using hierarchical stepwise linear regression. Basic characteristics, including age, sex, body mass index, self-perceived health status, type of institution, and program of study, were first entered (Model 1) as control variables followed by the six subscales of the Health-Promoting Lifestyle Profile II in the second step (Model 2). Finally, the Big Five personality traits, including the scores of openness to experience, conscientiousness, extraversion, agreeableness, and neuroticism, were added in the third step (Model 3). Multicollinearity in the independent variables of Model 3 was assessed using the variance inflation factor (VIF). Autocorrelation in residuals was evaluated using Durbin–Watson statistic. A value of 2 meant that there was no autocorrelation in the sample. A two-tailed *p* value < 0.05 was considered statistically significant.

## 3. Results

A total of 600 questionnaires were distributed, and 20 questionnaires had a missing response in the outcome variable or independent variables. Therefore, 580 participants were included in the final analysis.

The basic characteristics of the respondents are shown in Table 1. The mean age was 21.3 years, and a larger proportion of the respondents were female (79.3%). More than half of the respondents had a body mass index in the normal range (55.0%) and 21.2% of the respondents were underweight. More than half of the respondents reported that their perceived health status was average (50.9%). Regarding the type of institution, 66.2% of the respondents were students from technical universities, the remaining 33.8% were students from universities, and 31.2% were enrolled in health-, medical-, and social-welfare-related study programs. Regarding the negative emotional state, the mean score of the total DASS-21 score was 28.2 (SD 22.0) with a range from 0 to 122. The mean score of HPLP-II was 2.45 (SD 0.41). The mean scores for the Big Five personality traits were 3.22 (SD 0.64), 3.06 (SD 0.57), 3.02 (SD 0.71), 3.57 (SD 0.53), and 3.14 (SD 0.67) for openness to experience, conscientiousness, extraversion, agreeableness, and neuroticism, respectively.

Table 2 shows the results of the hierarchical linear regression analysis of the DASS-21 scores. The basic characteristics of the students, tested in Model 1, explained 4.7% of the variance in the DASS-21 scores. In Model 2, the six subscales of the HPLP-II were included as the independent variables, which explained an additional 13.0% (*p* < 0.001) of the variance in the DASS-21 scores. In Model 3, the Big Five personality traits were added, which further explained an additional 18.8% of the variance in the DASS-21 scores. In Model 3, neuroticism (standardized beta [std. β] = 0.450, *p* < 0.001) and openness to experience (std. β = 0.116, *p* = 0.003) were independently and significantly associated with the DASS-21 score. Agreeableness (std. β = −0.107, *p* = 0.007) was inversely associated with the DASS-21 score. In addition, age (std. β = 0.083, *p* = 0.018) was significantly associated with the DASS-21 score, while the score of the HPLP-II spiritual growth subscale (std. β = −0.216, *p* = 0.001) was inversely associated with the DASS-21 score. The VIF values for the variables in Model 3 ranged from 1.05 to 3.54, indicating an absence of multicollinearity.

## 4. Discussion

This present study investigated the association between negative emotional states and the five domains of personality in university students in Taiwan. The main finding was that neuroticism and openness to experience were significantly associated with negative emotional states, whereas agreeableness was inversely associated with negative emotional states. These associations remained significant when age and total score of HPLP-II were controlled for.

Neuroticism shows consistent and robust associations with mental disorders, especially depressive illness [36,37]. People who score high on neuroticism have a tendency to experience negative emotions, including feelings of sadness, anxiety, and anger [38]. A cross-sectional study of 575 medical students in China reported that neuroticism was strongly associated with personal distress (std. β = 0.53, *p* < 0.01). The authors suggested that the strong positive association between the two constructs could be explained by the negative emotionality and maladaptive emotion regulation that occurs in both neuroticism and personal distress [39].

Another study of 1738 Chinese undergraduate medical students also reported that neuroticism was positively related to depressive symptoms, while agreeableness and openness to experience were inversely related to the symptoms, after adjustment for age and sex. The present study also found that agreeableness was inversely associated with negative emotional states. However, students high in openness to experience were associated with unfavorable negative emotional states [40].

Individuals high in agreeableness are sympathetic and cooperative, which can be manifested as being likeable and harmonious in relations with others [38]. An online study of 635 Finnish university students revealed that students who scored high in agreeableness had a lower tendency for rumination, self-reported stress, depressive symptoms, and anxiety [41]. A Korean cross-sectional study also showed that low agreeableness was associated with depression in young adults [42]. The association between agreeableness and lower stress levels could be explained by the fact that these individuals tend to avoid interpersonal conflict and therefore experience less social stress [43,44].

People high in openness to experience are characterized by a wide range of interests and a recurrent need to experience variety and novelty. A laboratory study of 70 college students demonstrated that higher openness to experience was associated with better blood pressure adaptation with greater decreases in physiological reactivity in the two successive exposures to socially evaluative stressors [45]. In contrast, a study of 352 middle-aged adults showed that those with lower openness to experience had blunted and possibly maladaptive cortisol and cardiac responses to a laboratory psychological stress test [46]. Moreover, individuals with bipolar disorder are characterized by high openness to experience in addition to high neuroticism and low extraversion, agreeableness, and conscientiousness [47].

To date, the association between openness to experience and depression has been mixed [48]. A study showed that higher scores in the subsets of openness to experience were related to depression [49], while another study reported that individuals with major depression showed lower levels of openness to experience compared to controls [50]. A recent study of 531 members of the Old Order Amish and Mennonite (OOA/M) community in the United States showed that greater openness to experience was associated with greater symptoms of depression. In addition, openness to experience was also associated with stressful life events. The authors provided several possible explanations, and one was that new experiences might be in conflict with the conservative nature of the OOA/M culture, leading to increased stress [51]. Our finding was also in line with a cross-sectional study of 871 female Spanish university students, which showed openness to experience was significantly associated with an increased risk of major depression (odds ratio 1.08, 95% CI 1.02–1.14, *p* = 0.008) [52]. Although inconsistent results regarding the association between openness to experience and stress response deserve further investigation, it was suggested that the differences could be explained by a balance between two opposite effects of openness to experience: a higher level of sensitivity to new experiences on the one hand, and more efficient regulation of the hypothalamic–pituitary–adrenal stress responsivity on the other hand [53]. In addition, according to the Self-Awareness Theory of Reactive Depression, individuals with a high openness to experience might be more likely to experience a large discrepancy between their actual state and desired states, which could lead to a higher risk of depression [52]. Furthermore, it is possible that lockdowns during the COVID-19 pandemic period constrained the opportunities for seeking new experiences. A study based on the United Kingdom Household Longitudinal Study showed that openness was a strong predictor of mental health deterioration during the COVID-19 pandemic period [54].

In addition to the findings on the Big Five personality traits, this study also found that an older age was significantly associated with a higher score of negative emotional states. A study of 374 undergraduate students in the United States reported that the mean stress score measured by the DASS-21 was significantly higher in the upper years [55]. A cross-sectional study of 1738 medical students in China also reported that the depressive symptom score, as measured using the Center for Epidemiologic Studies Depression scale (CES-D), was significantly higher in students aged 22–28 years compared to those 15–21 years old [40]. The association between age and negative emotional states could be explained by the increasing workload in more senior years and uncertainties about their future as students approach graduation [56].

Another finding in this study was that the score of the spiritual growth subscale of the HPLP-II was significantly associated with a low score of negative emotional states. Spirituality can empower people when facing hardship and managing stress [57]. Previous research has documented an inverse association between depression and spirituality [58,59]. A cross-sectional study of 1276 nursing students in Taiwan found that spiritual health was inversely associated with clinical practice stress and depressive tendencies [60]. Another cross-sectional study of 500 Chinese university students also reported that university students with high spiritual wellbeing were also likely to experience fewer symptoms of depression, anxiety, and stress [61]. Future studies should confirm whether promoting spiritual growth can provide university students with a useful resource to manage negative emotional states.

Several limitations should be mentioned in this study. First, this study used a cross-sectional design and therefore the causal relationship between negative emotional states and the Big Five personality traits needs to be confirmed by prospective studies in the future. Second, all variables were assessed with self-reported questionnaires, which relied on the accuracy of the participant’s recall. Third, the internal consistency of the agreeableness subscale was not adequate with a McDonald’s omega value < 0.70. Fourth, potential influencing lifestyle factors, such as exercise and dietary habit, were not ascertained directly. Nevertheless, their potential confounding effects were adjusted using the HPLP-II in the multiple regression model.

## 5. Conclusions

The present study investigated the association between the Big Five personality traits and negative emotional states in university students. Our findings showed that neuroticism and openness to experience were positively associated, while agreeableness was inversely associated with negative emotional states. In addition, university students with a lower spiritual growth were associated with unfavorable negative emotional states. These students might benefit from targeted prevention and early intervention, which deserves investigation in future research.

## Figures and Tables

**Table 1 ijerph-19-16468-t001:** Characteristics of the respondents (N = 580).

Variable	Response Category	Frequency (%)
**Age, years, mean (SD)**		21.3 (1.0)
**Sex**		
	Male	120 (20.7)
	Female	460 (79.3)
**Body mass index category**		
	Normal	319 (55.0)
	Underweight	123 (21.2)
	Overweight	73 (12.6)
	Obese	63 (10.9)
	Missing	2 (0.3)
**Self-perceived health status**		
	Very good	88 (15.2)
	Good	173 (29.8)
	Average	295 (50.9)
	Poor	24 (4.1)
	Very poor	0 (0)
**Type of institution**		
	Technical university	384 (66.2)
	University	196 (33.8)
**Program of study**		
	Health-, medical-, and social-welfare-related	181 (31.2)
	Other	399 (68.8)
**Depression Anxiety Stress Scale-21 (DASS-21), mean (SD)**		28.2 (22.0)
**Health-Promoting Lifestyle Profile II, mean (SD)**		
Health responsibility		2.36 (0.53)
Physical activity		2.04 (0.57)
Nutrition		2.33 (0.44)
Spiritual growth		2.64 (0.57)
Interpersonal relations		2.78 (0.52)
Stress management		2.52 (0.50)
**Big Five personality traits, mean (SD)**		
Openness to experience		3.22 (0.64)
Conscientiousness		3.06 (0.57)
Extraversion		3.02 (0.71)
Agreeableness		3.57 (0.53)
Neuroticism		3.14 (0.67)

SD: standard deviation.

**Table 2 ijerph-19-16468-t002:** Hierarchical linear regression analysis of DASS-21 scores (N = 580).

Variable	Model 1			Model 2			Model 3		
	β	Std. β	*p*	β	Std. β	*p*	β	Std. β	*p*
**Intercept**	5.39	–	0.790	33.39	–	0.090	−29.13	–	0.136
**Age (years)**	0.68	0.031	0.460	1.21	0.055	0.165	1.82	0.083	0.018
**Sex**									
Male vs. female	1.29	0.024	0.595	1.00	0.018	0.667	1.65	0.030	0.426
**Body mass index**									
Underweight vs. normal	3.31	0.062	0.153	2.34	0.044	0.283	1.68	0.031	0.389
Overweight vs. normal	−0.47	−0.007	0.869	−0.42	−0.006	0.874	−0.08	−0.001	0.973
Obese vs. normal	5.60	0.080	0.065	5.02	0.072	0.078	3.16	0.045	0.212
**Self-perceived health status**									
Good vs. very good	0.64	0.013	0.823	−1.63	−0.034	0.544	−4.45	−0.092	0.062
Average vs. very good	7.41	0.168	0.006	1.72	0.039	0.503	−3.82	−0.087	0.098
Poor vs. very good	19.73	0.176	<0.001	10.16	0.091	0.038	1.06	0.009	0.810
**Type of institution**									
Technical university vs. university	1.91	0.041	0.416	2.20	0.047	0.318	0.71	0.015	0.717
**Program of study**									
Health-, medical-, and social-welfare-related vs. other	−4.22	−0.089	0.057	−4.32	−0.091	0.039	−3.43	−0.072	0.067
**Health-Promoting Lifestyle Profile II**									
Health responsibility				3.40	0.082	0.133	1.48	0.035	0.463
Physical activity				1.06	0.027	0.618	1.12	0.029	0.559
Nutrition				−1.40	−0.028	0.605	−1.16	−0.023	0.629
Spiritual growth				−15.23	−0.397	<0.001	−8.29	−0.216	0.001
Interpersonal relations				2.33	0.055	0.359	1.23	0.029	0.617
Stress management				−3.41	−0.078	0.181	−0.54	−0.012	0.816
**Big Five personality trait**									
Openness to experience							4.01	0.116	0.003
Conscientiousness							−1.70	−0.044	0.231
Extraversion							−0.44	−0.014	0.737
Agreeableness							−4.46	−0.107	0.007
Neuroticism							14.89	0.450	<0.001
Durbin–Watson	2.11			2.10			2.03		
Adjusted R^2^	0.047			0.170			0.358		
ΔR^2^	–			0.130			0.188		
*p*	–			<0.001			<0.001		

β, unstandardized regression coefficient; ΔR^2^, the change in R^2^ values from one model to another; R^2^, the proportion of explained variance in DASS-21 scores by the model; std. β, standardized regression coefficient.

## Data Availability

The datasets analyzed during the current study are available from the corresponding author on a reasonable request.

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
