# Peer review of "The Big Five Personality Traits as Predictors of Negative Emotional States in University Students in Taiwan"

_ijerph, 2022, doi:10.3390/ijerph192416468_

Round 1

Reviewer 1 Report

Dear authors,

thank you for your manuscript. I have some minor comments which I marked in the manuscript and some questions which I also wrote in the manuscript.

Best wishes

Author Response

We thank the reviewer for the comments and questions. We have made the changes suggested by the reviewer.

In addition, the reviewer asked “Could the openness to experience effect be attributed to the COVID pandemic? The search for new and varied stimuli could be extremely curtailed due to lockdowns (if there were ones in taiwan?) and this might lead to more negative emotional state..”.

We agree with the reviewer and have revised the manuscript accordingly and added a new reference (no. 54) (Proto, E.; Zhang, A. COVID-19 and mental health of individuals with different personalities. Proc Natl Acad Sci USA, 2021, 118, e2109282118.)

Reviewer 2 Report

This study sought to investigate the explanatory power of the big 5 personality traits, namely, Neuroticism, Extraversion, Openness to experience, Agreeableness, and Conscientiousness, on university studentsdistress. 

Introduction 

Comment 1. It would be quite interesting to know about the mental health status of university students in Western countries too. Please, report additional prevalence studies in this regard. Further, it might help to include more recent publications. 

Comment 2. Please, add information about previous studies of the relation between the big 5 and distress in this population and/or others. This is a research line that has been deeply studied. 

Methods and results 

Comment 3. The internal consistency of the Agreeableness subscale is not adequate (<.70). Please, add this as a limitation of the study. 

Comment 4. I disagree with the choice of the stepwise procedure. Stepwise regression has shown a fundamental problem, “some real explanatory variables that have causal effects on the dependent variable may happen to not be statistically significant, while nuisance variables may be coincidentally significant. As a result, the model may fit the data well in-sample, but do poorly out-of-sample.” (Smith, 2018). I strongly suggest to perform a hierarchical multiple regression with the sociodemographics and other questionnaires in the first step and the personality variables in the second one. Maybe the HPLP-II scores might be in an independent step (previous to the big 5). 

References 

Smith, G. Step away from stepwise. J Big Data 5, 32 (2018). https://doi.org/10.1186/s40537-018-0143-6

Author Response

Reviewer 2, Comment 1:

Introduction: It would be quite interesting to know about the mental health status of university students in Western countries too. Please, report additional prevalence studies in this regard. Further, it might help to include more recent publications. 

Response to Reviewer 2, Comment 1:

We thank the reviewer for the suggestion. We have added a description of the following four studies of university students in Western countries:

“A cross-sectional study of 1074 Spanish college students also reported a prevalence of 18.4%, 23.6%, and 34.5% for depression, anxiety, and stress, respectively [5]. A cohort study of 1686 undergraduate students in the United Kingdom found that 32% of students reported moderate to severe anxiety symptoms and 27% of students reported moderate to severe depressive symptoms at entry to university [6]. In addition, multiple studies in the United States have suggested there was an increasing prevalence of depression and anxiety among university students [7,8].” [Line 34-40]

New references:

  1. Ramón-Arbués, E.; Gea-Caballero, V.; Granada-López, J. M.; Juárez-Vela, R.; Pellicer-García, B.; Antón-Solanas, I. The prevalence of depression, anxiety and stress and their associated factors in college students. Int. J. Environ. Res. Public Health 2020, 17, 7001.

  1. Adams, K.L.; Saunders, K.E.; Keown-Stoneman, C.D.G.; Duffy, A.C. Mental health trajectories in undergraduate students over the first year of university: a longitudinal cohort study. BMJ Open 2021, 11, e047393.

  1. Lipson, S.K.; Lattie, E.G; Eisenberg, D. Increased rates of mental health service utilization by U.S. college students: 10-year population-level trends (2007-2017). Psychiatr. Serv. 2019, 70, 60–63.

  1. Oswalt, S.B.; Lederer, A.M.; Chestnut-Steich, K.; Day, C.; Halbritter, A.; Ortiz, D. Trends in college students' mental health diagnoses and utilization of services, 2009-2015. J. Am. Coll. Health 2020, 68, 41–51.

-------------------------------------------------------------

Reviewer 2, Comment 2:

Please, add information about previous studies of the relation between the big 5 and distress in this population and/or others. This is a research line that has been deeply studied. 

Response to Reviewer 2, Comment 2:

We agree with the reviewer that this is a research line that has been deeply studied, especially in adult populations. Nevertheless, few studies have investigated the Big Five personality and negative emotional states in Taiwanese university students.

We have revised the last paragraph of the Introduction section as:

“It has been associated with a wide range of behaviors in students, such as academic achievement [12], learning strategies [13], social well-being [14], and mental health [15]. Previous research has also demonstrated a number of associations between mental health and the five-factor model of personality in adults [16,17]. Similar associations were also observed in adolescents and students. For example, a higher level of Neuroticism is a risk-factor for depression and suicidal behavior [18,19]. A cross-sectional study of 323 Chinese undergraduates reported that Neuroticism, Conscientiousness, and Agreeableness significantly predicted anxiety among college students [20]. A study of 1744 students studying veterinary medicine, medicine, dentistry, pharmacy, and law in the United Kingdom showed that high levels of neuroticism and low conscientiousness were risk factors for increased psychological morbidity [21]. A six-year longitudinal study in Norway identified that a combination of high Neuroticism, high Conscientiousness, and low Extraversion could predict medical school stress [22]. A review study of 66 meta-analyses with 851 effect sizes revealed that while Neuroticism was consistently associated with common mental disorders, other traits also showed substantial independent effects [18].” [Line 49-64]

New references:

  1. Liu, W.; Lin, J. The role of meditation in college students' neuroticism and mental health. Transl. Neurosci. 2019, 10, 112–7.

  1. Lyon, K.A.; Juhasz, G.; Brown, L.J.E.; Elliott, R. Big Five personality facets explaining variance in anxiety and depressive symptoms in a community sample. J Affect. Disord. 2020, 274, 515–521.

  1. Ka, L.; Elliott, R.; Ware, K.; W., Juhasz, G.; Lje, B. Associations between facets and aspects of big five personality and affective disorders: A systematic review and best evidence synthesis. J. Affect. Disord. 2021, 288, 175–88.

  1. Lewis, E.G.; Cardwell, J.M. The big five personality traits, perfectionism and their association with mental health among UK students on professional degree programmes. BMC Psychol. 2020, 8, 54.

-------------------------------------------------------------------------

Reviewer 2, Comment 3:

The internal consistency of the Agreeableness subscale is not adequate (<.70). Please, add this as a limitation of the study. 

Response to Reviewer 2, Comment 3:

We appreciate the reviewer for this comment and we have added the following sentence in the limitation: “The internal consistency of the Agreeableness subscale was not adequate with a McDonald’s omega less than 0.70.” [Line 288-289]

-------------------------------------------------------------------------

Reviewer 2, Comment 4:

I disagree with the choice of the stepwise procedure. Stepwise regression has shown a fundamental problem, “some real explanatory variables that have causal effects on the dependent variable may happen to not be statistically significant, while nuisance variables may be coincidentally significant. As a result, the model may fit the data well in-sample, but do poorly out-of-sample.” (Smith, 2018). I strongly suggest to perform a hierarchical multiple regression with the sociodemographics and other questionnaires in the first step and the personality variables in the second one. Maybe the HPLP-II scores might be in an independent step (previous to the big 5). 

References 

Smith, G. Step away from stepwise. J Big Data 5, 32 (2018). https://doi.org/10.1186/s40537-018-0143-6

Response to Reviewer 2, Comment 4:

We greatly appreciate the suggestion from the reviewer. We have re-conducted the statistical analysis using hierarchical linear regression with the sociodemographics and other basic characteristics in the first step, HPLP-II scores in the second step, and the BIG Five personality variables in the third step. We have revised the Abstract [Line 19-22], Methods [Line 158-167], Table 2, and the Results [Line 187-194] sections accordingly. The results from the new hierarchical linear regression are similar to those based on stepwise multiple linear regression in the original version of the manuscript.

Reviewer 3 Report

It was a great pleasure that I reviewed the manuscript entitled “The big five personality traits as predictors of negative emotional states in university students in Taiwan.” I do not think the paper presented any significant findings. I am presenting only few comments to believe so in chronological order.

1.     Although the Introduction section is well-written mostly reviewing past findings related to the relationships between the big-five personality traits and negative affectivity, it is not clear why the authors wanted to conduct (or replicate) such well-reported studies again.

2.     In the section of Measurements, the authors should discuss why they also measured body weight and height for BMI. Also, they should describe how self-perceived health status was measured (although I could tell from Table 1).

3.     In the Results section, I do not think it is necessary for the authors to report all Ms and SDs for each of the big-five personality traits as they were reported in Table 1. 

4.     Most importantly, it is unclear how the authors executed the multiple stepwise regression analysis. They needed to report what variables were included in the 1st model, then what variables were included in the 2nd model, so on and so forth. Because the authors (under Table 2) also stated that other independent variables included all variables examined in this study and adjusted r-squared was pretty high, I think the authors actually performed the simultaneous multiple regression analysis instead. Either way, the analysis they conducted does not seem right.

Author Response

Reviewer 3, Comment 1:

It was a great pleasure that I reviewed the manuscript entitled “The big five personality traits as predictors of negative emotional states in university students in Taiwan.” I do not think the paper presented any significant findings. I am presenting only few comments to believe so in chronological order.

  1. Although the Introduction section is well-written mostly reviewing past findings related to the relationships between the big-five personality traits and negative affectivity, it is not clear why the authors wanted to conduct (or replicate) such well-reported studies again.

Response to Reviewer 3, Comment 1:

We thank the reviewer for the valuable comment. We agreed that the relationship between the big-five personality traits and negative affectivity has been studied in the past. Nevertheless, the association between Openness to experience and depression has been mixed. In addition, few studies have explored the relationship between personality traits and negative emotional states among university students in Taiwan. Therefore, we conducted the present study to further explore this relationship.

-------------------------------------------------------------------------

Reviewer 3, Comment 2:

In the section of Measurements, the authors should discuss why they also measured body weight and height for BMI. Also, they should describe how self-perceived health status was measured (although I could tell from Table 1).

Response to Reviewer 3, Comment 2:

“Body mass index was calculated in the present study because recent research has shown that it was associated to depression and psychosocial stress [23]. Self-perceived health status was measured using a single item with a five-point scale from very good to very poor.” [Line 99-101]

References:

  1. Eik-Nes, T.T.; Tokatlian, A.; Raman, J.; Spirou, D.; Kvaløy, K. Depression, anxiety, and psychosocial stressors across BMI classes: A Norwegian population study - The HUNT Study. Front. Endocrinol. 2022, 13, 886148.

-------------------------------------------------------------------------

Reviewer 3, Comment 3:

In the Results section, I do not think it is necessary for the authors to report all Ms and SDs for each of the big-five personality traits as they were reported in Table 1. 

Response to Reviewer 3, Comment 3:

We thank the comment from the reviewer. We included the means and standard deviations in Table 1 for completeness as these summary statistics were also provided by other variables.

-------------------------------------------------------------------------

Reviewer 3, Comment 4:

Most importantly, it is unclear how the authors executed the multiple stepwise regression analysis. They needed to report what variables were included in the 1st model, then what variables were included in the 2nd model, so on and so forth. Because the authors (under Table 2) also stated that other independent variables included all variables examined in this study and adjusted r-squared was pretty high, I think the authors actually performed the simultaneous multiple regression analysis instead. Either way, the analysis they conducted does not seem right.

Response to Reviewer 3, Comment 4:

We appreciate the reviewer’s comment regarding the statistical analysis. We have re-conducted the statistical analysis using hierarchical linear regression with the sociodemographics and other basic characteristics in the first step, the six subscales of the HPLP-II in the second step, and the Big Five personality variables in the third step. We have revised the Abstract [Line 19-22], Methods [Line 158-167], Table 2, and the Results [Line 187-194] sections accordingly. The results from the new hierarchical linear regression are similar to those based on stepwise multiple linear regression in the original version of the manuscript.

-------------------------------------------------------------------------

Round 2

Reviewer 2 Report

Thank you for answering my comments. The manuscript have significantly improved.

Reviewer 3 Report

Thank you for addressing most of my comments on previous submission.